# In Silico and In Vitro Screening of Antipathogenic Properties of *Melianthus comosus* (Vahl) against *Pseudomonas aeruginosa*

**DOI:** 10.3390/antibiotics10060679

**Published:** 2021-06-05

**Authors:** Itumeleng T. Baloyi, Idowu J. Adeosun, Abdullahi A. Yusuf, Sekelwa Cosa

**Affiliations:** 1Division of Microbiology, Department of Biochemistry, Genetics and Microbiology, University of Pretoria, Private Bag X20, Hatfield Pretoria 0028, South Africa; u18372882@tuks.co.za (I.T.B.); u21747050@tuks.co.za (I.J.A.); 2Department of Zoology and Entomology, University of Pretoria, Hatfield Pretoria 0028, South Africa; abdullahi.yusuf@up.ac.za

**Keywords:** anti-quorum sensing, antibiofilm, antivirulence, phytochemical compounds, plant extracts, GC-MS, molecular docking

## Abstract

Bacterial quorum sensing (QS) system regulates pathogenesis, virulence, and biofilm formation, and together they contribute to nosocomial infections. Opportunistic pathogens, such as *Pseudomonas aeruginosa*, rely on QS for regulating virulence factors. Therefore, blocking the QS system may aid management of various infectious diseases caused by human pathogens. Plant secondary metabolites can thwart bacterial colonization and virulence. As such, this study was undertaken to evaluate three extracts from the medicinal plant, *Melianthus comosus*, from which phytochemical compounds were identified with potential to inhibit QS-dependent virulence factors in *P. aeruginosa.* Chemical profiling of the three extracts identified 1,2-benzene dicarboxylic acid, diethyl ester, neophytadiene and hexadecanoic acid as the common compounds. Validation of antibacterial activity confirmed the same MIC values of 0.78 mg/mL for aqueous, methanol and dichloromethane extracts while selected guanosine showed MIC 0.031 mg/mL. Molecular docking analysis showed anti-quorum sensing (AQS) potential of guanosine binding to CviR’ and 2UV0 proteins with varying docking scores of −5.969 and −8.376 kcal/mol, respectively. Guanosine inhibited biofilm cell attachment and biofilm development at 78.88% and 34.85%, respectively. Significant swimming and swarming motility restriction of *P. aeruginosa* were observed at the highest concentration of plant extracts and guanosine. Overall, guanosine revealed the best swarming motility restrictions. *M. comosus* extracts and guanosine have shown clear antibacterial effects and subsequent reduction of QS-dependent virulence activities against *P.*
*aeruginosa*. Therefore, they could be ideal candidates in the search for antipathogenic drugs to combat *P.*
*aeruginosa* infections.

## 1. Introduction

*Pseudomonas aeruginosa* is an opportunistic Gram-negative bacterium (GNB) that causes chronic infections in humans, especially in immunocompromised individuals, and can be fatal in hazard to hospitalized patients [1,2]. *Pseudomonas aeruginosa* pathogenicity and colonization of on host tissues, surfaces, or medical devices cause nosocomial infections including urinary tract and wound infections, as well as cystic fibrosis [3]. This pathogen triggers biofilm formation, which is affected by biotic and abiotic factors and several virulence genes in a cell density-dependent manner through quorum-sensing (QS), a common communication system in bacteria [4].

Biofilms are defined as aggregates of bacterial cells that are enclosed in self-produced extracellular polymeric substances (EPS) [5]. Biofilm formation requires factors such as cell appendages, surface proteins, EPS, and cell motility regulated by second messengers of cyclic adenosine monophosphate (cAMP) and cyclic dimeric guanosine monophosphate (c-di-GMP). These play a significant role as a communication network to environmental factors involved in the formation of biofilm [6]. The EPS serves as a defensive shield for bacteria against antimicrobials, making biofilm eradication difficult [6]. Other factors that contribute to acute and chronic infections include virulence phenotypes such as pyocyanin, which aids in immune evasion, LasA protease, which disrupts the epithelial barrier, LasB proteases, which degrade matrix proteins, rhamnolipids, which cause immune cell necrosis, and LecA lectin, which promotes bacterial colonization. These factors are coordinated by quorum sensing system (QSS) in *P. aeruginosa* [7].

*P. aeruginosa* has four known QS systems namely, the LasI/LasR, RhlI/RhlR, *Pseudomonas* quinolone signaling (PQS) and integrated quorum signaling (IQS), with each utilizing unique autoinducers [3]. The Las system consists of a transcriptional activator (LasR) that activates LasI synthase thereby producing autoinducers *N*-(3-oxododecanoyl)-L-homoserine lactone (3-oxo-C12-HSL) [8]. While the Rhl system consists of RhlR activating RhlI synthase producing *N*-butyryl-L-homoserine lactone (C4-HSL) [9], the PQS system produces 2-heptyl-3-hydroxy-4-quinolone that is chemically unique from the autoinducer signal molecules of the Las and Rhl systems [10]. Lastly, the IQS structurally established 2-(2-hydroxyphenyl)-thiazole-4-carbaldehyde molecule [1]. These cell communication signals are interconnected and regulated hierarchically, triggering virulence factors and antibiotic resistance [11], thus necessitating novel approaches for combating *P. aeruginosa* infections.

Quorum sensing inhibition (QSI) strategies promise to combat multidrug resistant bacteria due to their ability to regulate pathogenicity and virulence [12]. Due to the growing understanding of the relationship between QSI and antimicrobial action, researchers are focusing on compounds that have the potential to disrupt the QS mechanism in pathogenic bacteria [2]. The GNB QS system uses acyl-homoserine lactones (AHLs) as signal molecules which are now targets and greatly studied in the discovery of antipathogenic agents/compounds [12]. The AHL molecular structure includes a homoserine lactone ring bonded to an acyl chain with an amide bond, and it plays a significant role in the pathogenicity of many GNB [13]. As a result, finding antivirulence or antipathogenic compounds that can target the bacterial QS mechanism regulating the virulence factors that mediate disease, is an alternative solution to antibiotic treatments [14].

Disruption of QS is a suitable strategy possibly due to its mechanism of action focusing on hindering the formation of biofilms and other virulence factors causing infection, without imposing selective pressure on the pathogen. Selective pressure is regarded as a key cause of bacterial resistance [2].

Medicinal plant decoctions have been investigated for their biologically active compounds as a source of antibacterial drugs since time immemorial. Currently, medicinal plants are receiving attention to aid a search for secondary metabolites with the potential to act against phytopathogens and prevent QS communication [15]. Secondary metabolites are an important source of antifungal, antibacterial, and antiviral agents [15]. However, little is known about their ability to disarm QS or as antipathogenic agents.

South Africa has a rich diversity of medicinal plants that are used as traditional remedies to treat various diseases in both humans and livestock [16]. *Melianthus comosus* (Vahl), a *Melianthaceae* has been commonly used as a remedy and treatment of ailments like gastrointestinal problems, respiratory problems, backache, snake envenomation, rheumatism, skin problems, septic wounds, and sores [17]. This plant contains glycosides, flavonoids, phytosterols, and triterpenoids, all of which exhibit several effects such as antibacterial, antifungal, anti-inflammatory, antioxidant, and cytotoxicity [16]. However, there is no report on QSI activities and in silico phytochemical screening of the plant.

The quest to find novel antipathogenic or QSI agents from medical plants is based on their phytochemistry containing promising classes of compounds including coumarins, terpenoids, benzoic acid derivates, flavonoids, and tannins [18]. The phytochemicals’ inhibitory mechanism depends on their structure, capability to inhibit competitively or not by binding to the substrate’s site other than the AHL, and having a structure that is similar to those of AHL molecule, which binds to the active site [19]. Docking describes ligand binding to a receptor by way of noncovalent interactions, which are frequently applied to examine the identification of new target compounds [20]. The recent approach focuses on the search for novel antipathogenic compounds utilizing a biomonitor strain of *Chromobacterium violaceum*, a well-accepted suitable organism to understand the molecular details of QS phenomenon in GNB. *Chromobacterium violaceum* produces QS-controlled purple pigment, violacein, and serves as a valuable model to comprehend the mode of action of many of the traditional medicinal products [7]. The QSS of this biomonitor strain consists of CviI/CviR, a LuxI/LuxR homologue. The CviI/CviR system uses autoinducer molecules such as *N*-hexanoyl-L-acyl homoserine lactone (C6-AHL) and *N*-decanoyl-L-homoserine lactone (C10-HSL) produced by autoinducer synthase CviI and binds to the CviR (receptor protein) to activate the expression of violacein production [21].

Here we provided an insight into the phytochemistry of *M. comosus* plant extracts using solvents of different polarities and validated their antibacterial and antipathogenic activities against *P. aeruginosa* using in silico and in vitro approaches. We employed molecular modeling due to its ability to provide proficient activity and binding affinities prediction of potential QSI compounds against protein active sites of test pathogens, *C. violaceum* and *P. aeruginosa*.

## 2. Results

### 2.1. Crude Extract Yields and Gas Chromatography–Mass Spectrophotometry (GC-MS) Profiling of Extracts

Comparatively, methanolic (22.1%) and aqueous (21.1%) extracts gave higher yields than ethyl acetate (3.6%) and dichloromethane extracts (2.2%).

The chemical profiles of the different plant extracts (aqueous, methanolic and dichloromethane) are shown in Figure 1. These compounds present in the three crude extracts belong to different chemical classes of hydrocarbons, organic acid esters, sugars, ketones, phytosteroids, acetates, and diterpene alcohols, with hydrocarbons and organic acid esters as major components. Full details of other key components are presented in Table 1, bolded numbers enclosed indicate peak numbers.

Aqueous extract of *M. comosus* contained 12 compounds with decane (1), guanosine (4), 1,2-benzene dicarboxylic acid, diethyl ester (6) and azelaic acid (9) as main components (Table 1) and the chromatograph (Figure 1). Further, 14 and 29 compounds were detected and identified in both extracts of *M. comosus* (methanol and dichloromethane), respectively, as presented in Table 1 and Figure 1. Four main constituents were identified in *M. comosus* methanol extract as decane (1), guanosine (4), 1,2-benzene dicarboxylic acid, diethyl ester (6) and neophytadiene (12). The main components detected from dichloromethane extract were heptacosane (29), nonacosane (32) dotriacontane (35) and hexatriacontane (36). The common compounds identified from all three extracts (aqueous, methanol and dichloromethane) were 1,2-benzene dicarboxylic acid, diethyl ester (6) and neophytadiene (12) and hexadecanoic acid (18).

### 2.2. In Silico Modeling of Identified Compounds against QS Receptors CviR’

Molecular docking studies were performed to investigate the binding potential and interactions of phytochemical compounds towards QS receptors: CviR’ and LasR protein. The structures of QS signal receptor CviR’ (PDB ID, 3QP1) from *C. violaceum* were used for the docking calculations. A cut-off docking score between −5 and −15 kcal/mol was established to differentiate between potential active and inactive compounds [22].

Figure 2 shows ligand-interaction of each ligand complex with the 3QP1 protein. The well-known QSI compounds such as quercetin and cinnamaldehyde were used as reference compounds to ascertain and gain insight into the binding mechanism with the QS proteins.

Molecular docking of the seven most dominant compounds from *M. comosus* extracts, with the receptor proteins, exhibited that compounds like sucrose (−7.591 kcal/mol), decanedioic acid (−6.997 kcal/mol), 1,2-benzene dicarboxylic acid, diethyl ester (−6.142 kcal/mol), and guanosine (−5.969 kcal/mol) were potential active QS compounds while the remainder weakly bound to the protein. Docking of sucrose to 3QP1 indicated that it is the most active potential QS compound with a docking score of −7.591 kcal/mol and glide energy of −32.204 kcal/mol; formed hydrogen bonds with Tyr80, Trp84 and Asp97, followed by decanedioic acid with a docking score of −6.997 kcal/mol and glide energy of −39.294 kcal/mol; formed hydrogen bonds with Ser155 and Tyr80. A 1,2-benzene dicarboxylic acid, diethyl ester compound also showed a docking score of −6.142 kcal/mol and glide energy of −33.463 kcal/mol; formed hydrogen bonds with Trp84, Ser 155 and Tyr80, followed by guanosine with a docking score of −5.969 kcal/mol and glide energy of −25.641 kcal/mol; formed hydrogen bonds with Trp84, Met135. Based on molecular docking study, compounds like phytol, neophytadiene and 2.3-dihydro-3.5-dihydroxy-6-methyl-4H-pyran-4-one showed the weaker docking score with the 3QP1 protein.

In contrast, the positive controls of quercetin and cinnamaldehyde showed docking scores of −10.613 and −5.949 kcal/mol, respectively. Quercetin indicated to bind to the active site interacting with 18 hydrophobic interactions. The residues involved in 3QP1-quercetin hydrogen bond interaction were Ser155, Asp97, Met135, π-π staking with Tyr80 and Tyr 88. Whereas, cinnamaldehyde displayed to bind to Trp84.

### 2.3. In Silico Modeling of Identified Compounds against LasR Protein

The studied compounds were evaluated for their ability to bind the structures of LasR (2UV0) protein, in order to validate the in vitro findings and identify possible compounds that could be responsible for the efficacy of plant extracts against *P. aeruginosa* with their mode of action. The binding residues are presented in Figure 3. The compounds of sucrose and guanosine presented the highest docking scores to 2UV0 with −10.424 and −8.376 kcal/mol, respectively, while quercetin showed −9.946 kcal/mol and 3-oxo-C12-homoserine lactone (signaling molecule) showed −9.803 kcal/mol docking scores.

The following compounds azelaic acid (−6.581 kcal/mol), 1,2-benzene dicarboxylic acid, diethyl ester (−5.285 kcal/mol) 2.3-dihydro−3.5-dihydroxy-6-methyl-4H-pyran-4-one (−5.358 kcal/mol) and phytol (−5.664 kcal/mol) also displayed notable docking scores. Guanosine revealed the highest glide energy at −49.614 kcal/mol followed by azelaic acid at −42.931 kcal/mol while reference compounds of quercetin cinnamaldehyde and 3-oxo-C12-HSL showed glide energies at −45.128, −24.305 and −61.317 kcal/mol, respectively. Both guanosine and azelaic acid were shown to have better glide energy as compared to our reference compound cinnamaldehyde.

The compound of sucrose was shown to have 13 hydrophobic interactions and formed hydrogen bonds with Thr75, Val76, Leu125 whereas guanosine formed hydrogen bonds with Tyr47, Arg61, Thr75, π-π staking with Tyr64 and 14 hydrophobic interactions. Azelaic acid compound was shown to form hydrogen bonds Tyr56, Ser129 and have 19 hydrophobic interactions while 2.3-dihydro-3.5-dihydroxy-6-methyl-4H-pyran-4-one formed 10 hydrophobic interactions and formed hydrogen bonds with Tyr56, Ser129, Trp60.

The reference compounds of quercetin, cinnamaldehyde and 3-oxo-C12-HSL appeared to bind to the active site of the 2UV0 protein. Quercetin was shown to form hydrogen bonds with Ser155, Asp97 and Met135, π-π staking with Tyr88 and Tyr 80, while cinnamaldehyde formed hydrogen bonds with Tyr56, Ser129 and have 10 hydrophobic interactions. A 3-oxo-C12-homoserine lactone formed hydrogen bonds with Ser129, Tyr56, Trp60, Asp73. The two reference compounds of cinnamaldehyde and 3-oxo-C12-homoserine lactone all formed hydrogen bonds with Ser129.

### 2.4. In Vitro Antibacterial Validation of M. comosus Crude Extracts against P. aeruginosa

Antibacterial activities of *M. comosus* crude extracts against *P. aeruginosa* showed MIC values ranging from 0.78 to 6.25 mg/mL (Table 2). *Melianthus comosus* aqueous, methanol and dichloromethane extracts showed the same MIC value of 0.78 mg/mL exhibiting noteworthy activity. Acetone extract showed MIC value of 1.56 mg/mL, while the ethyl acetate extract showed a higher MIC value of 6.25 mg/mL. The positive control, ciprofloxacin, showed significant MIC value of 0.001 mg/mL while negative control, 1% dimethyl sulfoxide (DMSO) showed no inhibitory effect (6.25 mg/mL) against *P. aeruginosa.*

Guanosine, a compound present in *Melianthus comosus* was selected for in vitro assays due to its revealed potent potential QSI results in *P. aeruginosa* and *C. violaceum*. The compound showed an MIC value of 0.031 mg/mL while quercetin (the positive control) showed 0.008 mg/mL MIC value (Table 2).

### 2.5. In Vitro Validation of Quorum Sensing-Dependent Violacein Inhibition (QSI)

*Chromobacterium violaceum*, a well-accepted QSI biomonitor strain which produces the purple pigment (known as violacein) was selected, as it is an excellent bacterium to visually detect and quantify pigment inhibition by phytochemicals. *M. comosus* (aqueous extract) exhibited AQS activity only, with an opaque zone of inhibition of 13, 14 and 15 mm at concentrations of 1.56, 3.12 and 6.25 mg/mL, respectively. While methanolic extract exhibited bactericidal activity only, showing clear zones of inhibition of 13 and 15 mm at the same concentrations. None of the other crude extracts exhibited the ability to inhibit QS-dependent violacein.

Cinnamaldehyde (positive QS inhibitor, as per documented literature) displayed antibacterial activity in a concentration-dependent manner with clear zones of inhibition of 13, 19, 13 and 26 mm at different concentrations of 0.5, 0.25, 0.12 and 0.06 mg/mL, respectively.

Based on the above qualitative QS-dependent violacein inhibition activity, not all the crude extracts demonstrated activity. Hence, the crude extracts as well as guanosine were subjected to quantitative violacein inhibition assay to further validate the violacein inhibitory effect and determine the exact concentration required for bactericidal or QS inhibition. Only the aqueous extract showed violacein inhibition at MIC (38.4%) and ½ MIC (24.3%) while the methanolic and dichloromethane extracts showed no inhibition (Figure 4).

Guanosine showed varying inhibitory effects at MIC, ½ MIC, ¼ MIC and ⅟₈ MIC with 40.24, 33.61, 27.19 and 26.00 percentage inhibitory concentrations, respectively. The positive control, quercetin also displayed violacein inhibitory activity at 52.54% (MIC), 48.61% (½ MIC), 45.68% (¼ MIC) and 41.50 (⅟₈ MIC) as shown in Figure 4.

The IC_50_ values of aqueous *M. comosus* crude extract and guanosine that were compared to those of the positive controls are shown on Table 3. Aqueous *M. comosus* displayed AQS activity with violacein inhibition of 20% at the lowest concentration of 0.78 mg/mL with an IC_50_ value of 1.52 mg/mL indicating a degree of their activity (Table 3). Cinnamaldehyde reduced violacein production activity exhibiting more than 40% inhibition at the lowest concentration (0.048 mg/mL).

### 2.6. P. aeruginosa Biofilm Formation Inhibition: Cell Attachment and Biofilm Development

The three crude extracts (aqueous, methanol and dichloromethane) did not inhibit biofilm but rather enhanced growth, while the positive control, ciprofloxacin inhibited cell attachment at 52.88% (ANOVA GLM, F = 182.77, DF = 4, R2 = 0.986, *p* < 0.05), (Table 4). Biofilm development inhibitory activity for ciprofloxacin was at 39.43% against the bacterium with differences found between the plant extracts and positive controls (ANOVA GLM, F = 2.21, DF = 4, R2 = 0.469, *p* < 0.05). Guanosine inhibited cell attachment at 78.88% while quercetin inhibit at 55.14%. Inhibition of biofilm development for guanosine and quercetin were at 34.85% and 44.35%, respectively (Table 4).

### 2.7. Inhibitory Effect of Plant Extracts and Compound on Pyocyanin Production

The three crude extracts (aqueous, methanol and dichloromethane) at different concentrations (2 x MIC, MIC, ½ MIC and ¼ MIC) showed varying pyocyanin inhibitory activity. The highest concentration of 1.56 mg/mL (2 x MIC) showed the strongest inhibition on the production of pyocyanin (OD value: 2.04) for dichloromethane extract and the same OD value of 2.22 for aqueous and methanol extracts while the lowest concentration 0.195 mg/mL (¼ MIC) showed the weakest pyocyanin inhibition for aqueous and methanolic extracts at OD values of 3.41 and 3.24, respectively. Methanol extract however showed least inhibition at ½ MIC (OD value: 2.56) (Figure 5).

Guanosine showed highest pyocyanin inhibition at both 2 x MIC (0.062 mg/mL) and MIC (0.031 mg/mL) at OD value of 2.39 while the least inhibition was at ½ MIC and ¼ MIC (OD value: 2.73). The positive control (ciprofloxacin) showed pyocyanin inhibition at OD value: 2.04 while untreated cell showed lesser pyocyanin production at value of 2.90 (Figure 5).

### 2.8. Effect of Plant Extracts and Compound on Swimming Motility

The swimming motility restriction was mostly observed at the highest concentration of plant extracts and guanosine as presented in Figure 6.

In contrast to untreated cells (10 mm), the aqueous extract showed only a small restriction of motility at 2 x MIC with a 9 mm zone diameter, while the methanol extract only decreased motility at MIC (9 mm) concentrations. As shown in Figure 5, dichloromethane extract of *M. comosus* decreased swimming motility at all concentrations measured. Guanosine, on the other hand, inhibited swimming motility at 2 x MIC and MIC concentrations, with swimming zone diameters of 7 and 8 mm, respectively, as compared to the untreated cell, which had a swimming zone diameter of 10 mm.

The three extracts and guanosine revealed swimming motility zones that can be compared with ciprofloxacin (positive control) showing zone diameter of 9 mm (Figure 6).

### 2.9. Effect of Plant Extracts and Compound on Swarming Motility

All extracts of *M. comosus* at the four tested concentrations showed reduced swarming motility, although at varying diameter zones when compared to the untreated cells (14 mm, Figure 7). The highest level of inhibition was recorded at 2 x MIC concentration for all extracts and guanosine (Figure 7).

Overall, guanosine revealed the best swarming motility restrictions particularly at 2 x MIC concentration (8 mm zone diameter) which is comparable with ciprofloxacin, the positive control that showed significant swarming motility limitation (7 mm zone diameter).

## 3. Discussion

Medicinal plants are an important source of medication for the global healthcare system, with approximately 80% of the population [23] depending on their use in conventional medicine, sparking interest in the discovery of effective plant extracts and secondary metabolites used in the management of microbial diseases [24]. The use of in silico molecular docking offers an efficient strategy to rapidly identify individual phytochemical compounds as potential QSIs that may be lead compounds for drug development. The strategy provides specificity since it employs potential compounds as antagonists against the active sites of QS-associated proteins.

*Melianthus comosus* (the medicinal plant of interest studied) extracts were obtained, extracted using solvents of different polarities. The solubility of plant extracts depends on the solvents used as each plant part has a variety of phytoconstituents. Different solvents have the potential to selectively extract antipathogenic compounds due to their polarities. Guanosine compound derived from *M. comosus* was also examined for its antipathogenic properties. Here, we showed that aqueous and methanol extracts had more phytochemicals compared to acetone, dichloromethane, and ethyl acetate. Congruent to our study, Eloff, Angeh, and McGaw, [17] and Mabona et al. [25] reported the highest percentage extraction yields from methanol and water; also stated that this indicates that *M. comosus* contains a higher concentration of polar compounds.

Analysis was carried out to determine the chemical profiles of *M. comosus* extracts using GC-MS. The profiles confirm the presence of phytoconstituents of pharmaceutical value and provide a better understanding of the nature of medicinal properties of the plants [26]. The GC-MS (Figure 1) analysis of *M. comosus* (aqueous, methanol and dichloromethane) revealed various classes of compounds such as flavonoids, phytosterols and terpenoids that possess biological activities which have a defense mechanism against pathogenic *P. aeruginosa* [17].

Based on the chemical profiles of these extracts and virtual molecular docking screening, it is feasible to explore the QS mechanisms to better understand the network of multiple responses prompted by AHLs [13].

Our molecular docking analyses revealed the potential of the identified compounds to disrupt the QS systems of both *C. violaceum* and *P. aeruginosa*. The C10-HSL ligand was re-docked into the active site of the CviR protein to check the feasibility of the docking protocol. Based on the results, compounds of 2.3-dihydro-3.5-dihydroxy-6-methyl-4H-pyran-4-one, neophytadiene, and phytol demonstrated weaker binding affinities as demonstrated by docking scores. Only sucrose, decanedioic acid, 1,2-benzene dicarboxylic acid, diethyl ester, and guanosine revealed good binding affinities to the CviR’ protein. This may be attributed to the compounds’ ability to fit in the AHL receptor’s pocket due to their structural similarity. As mentioned above, the best suitable QSI molecules depend on the compound’s structure and the capability of noncompetitive or competitive inhibition by targeting the active site [19]. Numerous studies have documented a range of plant compounds and their abilities to bind to the CviR protein’s active site using in silico molecular docking approach [27,28]. The QSI assay has shown that *M. comosus* extracts could disrupt the QS system of the biomonitor strain. However, extracts possess every single compound, and it cannot be affirmed that these compounds incited the QSI activity. Further studies are required to test each of the compounds to validate these findings.

*P. aeruginosa* was studied based on its QSS and the bacterium’s potential to target for the development of novel compounds with AQS activity. *P. aeruginosa* is activated by 3-oxo-C12-HSL to bind to the 2UV0 protein. Based on the results, most of the compounds such as guanosine followed by phytol and 2.3-dihydro-3.5-dihydroxy-6-methyl-4H-pyran-4-one including positive controls revealed improved docking scores.

Docking results of the compounds on the LasR receptor also revealed varying glide energy. High binding/glide energy scores of the compounds represents supportive energy for the protein–ligand binding interaction and indicates that they may be biologically active as well as highly efficient compounds [29].

Sucrose displayed a higher docking score as compared to the reference compounds whereas guanosine exhibited a docking score closer to both quercetin and 3-oxo-C12-HSL while cinnamaldehyde had lower scores though reported in literature as QSI agent, in vitro. Similar to our study, Kumar et al. [30] demonstrated that 3-oxo-C12-HSL showed dock scores of −9.0 kcal/mol for the 2UV0 protein. A report by [2] confirmed the results obtained here with quercetin molecule to compete with the natural ligand to bind to the LasR protein. These findings also revealed that 2.3-dihydro-3.5-dihydroxy-6-methyl-4H-pyran-4-one and 3-oxo-C12-HSL were both shown to form a hydrogen bond with Ser129, Tyr56 and hydroxyl group with Trp60.

Results from the in silico study also revealed interpretation of structure–function relationships amongst the tested compounds and QS inhibitors, which supports their potential use in combating bacterial virulence and QS activity. Guanosine exhibiting a docking score closer to quercetin can be attributed to the presence of functional groups common to both structures. They both contain a hydroxyl group, a functional group with the chemical formula -OH that is made up of one oxygen atom covalently bound to one hydrogen atom. Additionally, guanosine is composed of guanine which has a carbonyl group, a functional group composed of a carbon atom double-bonded to an oxygen atom: C = O which is also present in the structure of 3-oxo-C12-HSL and quercetin. These structural similarities can pose the similarity in their docking scores.

Although compounds such as azelaic acid, 1,2-benzenedicarboxylic acid, 3-dihydro-3,5-dihydroxy-6-methyl-4H-pyran-4-one and decanedioic acid showed potentials as quorum quenching compounds based on their docking scores, they did not show good binding affinity to our test pathogen *P. aeruginosa* (LasR: 2UV0) protein. Again, sucrose displayed a high docking score to both proteins (CviR and LasR), however, was excluded due to negative connotation of sugar as a potential drug candidate, hence was not considered for in vitro studies.

Guanosine was found to potentially act as antagonist of CviR and 2UV0 protein, the compound was thus chosen for further in vitro studies together with the crude extracts. Guanosine and its analogs are reported as antiprotozoal and antiviral agents [31].

When validating the antibacterial activities of the test plant extracts, significant minimum inhibitory concentrations (MICs) were observed for aqueous, methanol and dichloromethane crude extracts of *M. comosus* (Table 2). *Melianthus comosus* (aqueous and methanol) extracts showed a noteworthy MIC value of 0.78 mg/mL. van Vuuren and Muhlarhi [32] stated that a noteworthy activity is regarded as MIC values below 1 mg/mL. Contrary to our findings, Mabona et al. [25] reported higher MIC values of 2.00 mg/mL for aqueous extract of *M. comosus* against *P. aeruginosa* whereas Kelmanson et al. [33] reported no antibacterial activity for methanolic extracts of *M. comosus*. The improved antibacterial activity of *M. comosus* extracts was observed and could be due to our extraction and preparation methods where we air-dried collected leaves and extract at temperatures below 60 °C to prevent degradation or loss of active phytochemicals. Additionally, the geographic location of the plant could influence its phytochemistry and in turn affects its potency [34]. Results of the antibacterial activity of guanosine tested against *P. aeruginosa*, however, revealed a minimum inhibitory concentration of 0.031 mg/mL while quercetin, being the positive control revealed 0.008 MIC. The low MIC values suggests that a low concentration of the compounds is required for inhibiting the growth of the organism, hence drugs/compounds with lower MIC scores are more effective antimicrobial agents [35]. According to Gibbson et al. [36] in Mamabolo et al. [37], MIC values lower than 1 and 0.064 mg/mL are considered significant for crude plant extracts and individual phytochemicals, respectively. For this reason, both our extracts and guanosine showed potent activities and qualify as potential antibacterial agents against the studied test pathogen.

Due to the antibacterial potency of *M. comosus* extracts and activity of guanosine against *P. aeruginosa*, this study focused on the bioactive components of the plants and investigated these further.

Targeting antivirulence or anti-quorum sensing (AQS) presents a novel approach suitable as an alternative to bacterial killing through antibiotics to circumvent increasing antibiotic resistance presented in bacteria [38]. *Chromobacterium violaceum*, a biomonitor strain of choice for AQS screening, produces violacein pigment, encoded by the *vio* operon, whose expression is regulated by QS. The QS-regulated pigment unlocks the opportunity of discovering antivirulence compounds and/or plant extracts as it simplifies the visualization and quantification of violacein production [14]. While purple pigmentation inhibition in *C. violaceum* is a readily observable phenotype that aids in AQS screening, it does not reveal the exact types and numbers of active chemical compounds present. Signal binding, degradation, or direct interference with the gene are all strategies for achieving QSI [39].

Using the disc diffusion assay to qualitatively observe the violacein inhibition, only aqueous extracts of *M. comosus* presented potential by showing opaque zones of inhibition in a concentration-dependent manner. Congruent to this study, Chenia [21] reported QS inhibition opaque zones of *Kigelia africana* extracts in a concentration-dependent manner (0.31–8.2 mg/mL). A similar trend was reported by Baloyi et al. [38] where out of 70 plant extracts tested, only methanolic extracts of *Hydnora africana* exhibited opaque zones of inhibition in a dose dependent manner.

Similarly to our study, it was noted that most plant extracts qualitatively exhibited no AQS inhibition. However, that did not eliminate the screening of plant extracts further for quantitative AQS assay. Cosa et al. [40] suggested that agar well or disc diffusion assay might be an unsuitable method to assess AQS activity. Therefore, quantitative assays adopted for studies should be accepted and optimized methods, to exclude inconsistencies between two commonly used methods.

Quantitative AQS assay was used to determine the extent of violacein reduction at various concentrations by plant extracts. Based on the results, quantitative AQS activity demonstrated a concentration-dependent inhibitory effect of the plant extracts. Vasavi [41] reported a maximum of 80% violacein reduction of *Syzygium cumini* and *Pimenta dioica* extracts at 0.5 and 1 mg/mL, respectively. Ganesh and Rai [42] have also found *Terminalia bellerica* extract to reduce violacein production of *C. violaceum* by 66% inhibition at a concentration of 0.5 mg/mL. Hypothetically, antipathogenic drugs should exhibit their AQS potential with their efficacy demonstrated at sub-MIC concentration. Our current findings provide an insight into QS inhibitory effects of *M. comosus* extracts against the biomonitor strain in a dose-dependent manner. A novel approach in attenuation of bacterial resistance is by targeting the QSS that regulates the pathogenicity of bacteria [2].

The findings of this study indicate that guanosine, a plant-derived compound, is capable of inhibiting QS development, which is one of the reasons why plants have been used in traditional medicine. The quantitative AQS results of guanosine revealed as high as 40.24 percentage of violacein inhibition at 0.75 mg/mL, which is quite comparable with the result of quercetin, the positive control that had 52.54 as its highest percentage of violacein inhibition at 0.75 mg/mL. This may be due to a similarity in the structure of guanosine with the AHL molecule, enabling structural analogues of signaling molecules (AHL) to competitively bind with corresponding receptor protein, influencing the transmission of signal molecules and quorum sensing. According to our findings, guanosine, a purine nucleoside composed of guanine attached to a ribose (ribofuranose) ring through a beta N9 glycosidic bond, is a promising compound with the potential to inhibit biofilm formation and QS activity. Guanosine seems to be interesting as well because it is a purine nucleoside thought to have neuroprotective properties [43]. Furthermore, when guanosine is phosphorylated, it can be converted into cyclic guanosine monophosphate, a secondary messenger that is needed for many physiological processes including visual transduction, gene expression, and metabolic function. On the other hand, quercetin is known for its pharmacological effects, inhibition of biofilm and ultimately, inhibition of quorum sensing as previously described by Thi et al. [44].

Microorganisms such as *P. aeruginosa* interact via QSS for coordinated biofilm formation of biofilms, persistence, and production of other virulence factors [24]. Hence, *M. comosus* potential extracts were evaluated for biofilm formation and development disruption in *P. aeruginosa*. Our findings indicate that *P. aeruginosa* is more resistant to the three *M. comosus* plant extracts, as no inhibitory effect was observed; instead, an increased biofilm was observed. In contrast, a noteworthy antibacterial activity was observed for extracts made from *M. comosus*. One possible explanation is the resistance mechanism involved, especially during the cell attachment stage, which could be due to improved efflux pump action that expels plant extracts from the cells. This could also be because *P. aeruginosa* anchors tightly to the surface of the wells, requiring a higher concentration of crude extracts to disperse. Alternately, the plant extracts possibly possess additional nutrients for bacterial growth. However, numerous studies have shown the ability of different plant extracts to reduce *P. aeruginosa* biofilm formation [42,45]. Sarkar et al. [46] have reported that methanolic extract of *Kalanchoe blossfeldiana* inhibits the preformed biofilm of *P. aeruginosa* with 72.82% percentage inhibition at 1.25 mg/mL. Further, *Rosa rugosa* tea polyphenol has been tested against *P. aeruginosa* (PA01) and shown percentage inhibition of 72.90% at a concentration of 0.64 mg/mL [45]. Thus said, several other studies have proven that the eradication of biofilms is rather difficult as resistance was shown by various biofilm-forming microbes [46].

The antibiofilm activity of guanosine and quercetin were at 34.85% and 44.35%, respectively, suggesting that these compounds can inhibit the development of biofilms and attachment of cells to surfaces. The relevance of our results in evaluating the antibiofilm activity of quercetin was to demonstrate that this molecule, even at sub-inhibitory concentrations, has the ability to inhibit the formation of biofilm, which is consistent with findings presented by Costa Júnior et al. [47]. Quecan [28] also recorded for quercetin (0.032 mg/mL) a significant reduction in QS-dependent phenotypes including violacein production, biofilm formation, among other activities, in a concentration-dependent manner. Result from this study also revealed 78.88% inhibition of biofilm cell attachment by guanosine. Guanosine’s notable activity indicates its potential to rapidly invade target cells before the biofilm matures and disperses, implying that it may interfere with bacteria cell communication machinery.

This study envisaged that plant extracts could prevent the formation of biofilm and therefore considered the alternative approach of inhibiting some extracellular factors in *P. aeruginosa* such as bacterial motility (swimming and swarming) and pyocyanin production. Pyocyanin, a blue secondary metabolite capable of producing free radicals, is one of the main virulence determinants of *P. aeruginosa.* This compound acts by interfering with ion transfer and mucus secretion in respiratory epithelial cells, which is usually seen in cystic fibrosis patients [48]. Pyocyanin synthesis is often controlled by a complex synchrony of QSS including the lasR-lasI, rhlR-rhlI, and PQS systems, which affect the development of rhamnolipids, proteases, and elastases [49]. Results of the pyocyanin assay revealed varying inhibitory activity of the three crude extracts at different concentrations. The strongest pyocyanin inhibition was observed at the highest concentrations of the aqueous, methanol and dichloromethane extract which suggests a correlation between concentration and the extent of inhibition. The degree of pyocyanin inhibition by extracts and guanosine can be compared to the positive control with a slight difference with OD value of 0.18 against the untreated bacterium. In a study carried out by [50], pyocyanin level in *P. aeruginosa* treated with five plant extracts was also significantly reduced in contrast to the green pigment of untreated cultures. This could be explained as quorum-control of pyocyanin production. Furthermore, quorum quenching agents have a significant effect on the release of pyocyanin by *P. aeruginosa* [50].

Another reason for *P. aeruginosa* to be regarded as a major life-threatening opportunistic pathogen is its ability to colonize other environments through motility [48]. QS regulates motility in *P. aeruginosa*, including swimming on soft surface and swarming on semisolid surface, which are facilitated by flagella and pili IV [51]. The effects of *M. comosus* extracts and guanosine on the motility of *P. aeruginosa* were investigated in this study. We found that they exhibited a distinct influence on swimming and swarming motility compared to the negative control (untreated bacterial cells) (Figure 4). Plant extracts and guanosine treatment revealed a slight reduction in the swimming and swarming motility zones which corroborates the findings of Cosa et al. [3] where *Calpurnia aurea* extracts reduced swimming and swarming motility in *P. aeruginosa*. Additionally, in a study carried out by Lakshmanan et al. [52], the methanol extracts of *A. officinarum* and *C. tamala* significantly inhibited the swarming motility *of P. aeruginosa* when compared to untreated control. The swarming motility results were better than the swimming results obtained in our study which is also in tandem with the submission of Lakshmanan et al. [52]. Reduction in swarming area as compared to the control may indicate the presence of QSI compounds [39].

Interestingly, although the in silico results displayed good binding affinities of the individual compounds to 2UV0 protein, the antibiofilm activity exhibited by *M. comosus* extracts showed incapability of disrupting biofilm formation. This could be attributed to the plant compounds in extracts having an antagonistic effect on each other [53]. Conversely, *P. aeruginosa* uses four QS system and this could be due to extracts targeting the rhl system where RhlR controls the expression of genes required for biofilm formation instead of the Las system. Hossain et al. [54] stated that *P. aeruginosa* has a complex QS regulated system and could be attributed to circumstances where a similar compound hinders one virulence factor, and on the other hand promotes activation to the other or completely reducing the compound’s therapeutic potential. Therefore, future studies are required to elucidate the molecular mechanism of the compounds to act as antipathogenic drugs. Additionally, further research needs to be carried out on the in-depth study of molecular mechanism of the four QSS in *P. aeruginosa* at the same time.

Therefore, the in silico results demonstrated that the best compounds with QS potential are sucrose, 1,2-benzene dicarboxylic acid, diethyl ester and guanosine, which binds to both 3QP1 and 2UV0 protein. Quercetin was a better reference compound that bound to both CviR and LasR receptors with good binding affinities. To the best of our knowledge, this is the first study on *M. comosus* AQS activities and quorum quenching potential on CviR’ and 2UV0 proteins using in silico approach.

## 4. Materials and Methods

### 4.1. Plant Collection, Preparation, and Extraction

The leaves of *M. comosus* (Honey flower) medicinal plant was collected at Manie van der Schijff botanical (25° 45′ 15.84″ S, 28° 13′ 43.2084″ E) garden at the University of Pretoria, South Africa. The plant was positively identified by Ms Magda Nel from the Department of Plant and Soil Sciences, University of Pretoria and voucher specimens were deposited at the H.G.W.J. Schweickerdt herbarium of the University of Pretoria, South Africa with the following voucher numbers PRU 125457 for *M. comosus*. The plant was selected based on its medicinal uses in treating infections such as skin disease, respiratory tract infection and septicemia caused by *Enterobacteriaceae* bacteria especially *P. aeruginosa* bacterium.

After collection, the plant was extracted following a similar approach described by Adonizio [26], with slight modifications briefly as follows. The leaves were washed and left to air-dry at room temperature for 3–7 days. Thereafter, the dried leaves were ground into a fine powder using IKA MF 10.1 cutting grinder (Cole-Parmer scientific experts, Chicago, IL, USA)). In total, 25 g of the fine powder was added to 250 mL of the following solvents with varying polarities; methanol, acetone, ethyl acetate and dichloromethane and extracted for 48 h with agitation at 150 rpm using a 261-orbital shaker (Labotec, South Africa). After 48 h the extracts were filtered through a Whatman no. 1 filter paper (11 µm). The filtrates were dried using a Buchi rotary evaporator-interface I-100 (Labotec, Johannesburg, South Africa) at 50 °C. The collected crude extracts were transferred into 10 mL glass vials further dried in a fume hood cabinet and stored at 4 °C until required for further analysis. Aqueous extracts were made in a similar way using 25 g of powdered plant material added to sterile distilled water (250 mL) and boiled for 15 min then allowed to cool before filtering. The filtrates were transferred to glass bottles and lyophilized using a freeze-drier (SP Scientific, New York, NY, USA). All the dried extracts were weighed using a weighing balance (Kern 770, Microsep, Johannesburg, South Africa), re-dissolved in 1% aqueous dimethyl sulfoxide (DMSO) by sonication in a 701 ultrasonic bath (Labotec, Johannesburg, South Africa) and transferred to 10 mL vials. The stock solutions (100 mg/mL) for the 10 plant extracts were later diluted to the required concentrations (25 and 1 mg/mL) for the biological assays.

The percentage yields for each extract were calculated using Equation (1).
Percentage yield (%) = (dry crude extract/dry initial material before extraction) × 100(1)

### 4.2. Identification of Phytochemical Compounds Using Gas Chromatography–Mass Spectrophotometry (GC-MS)

Extracts were analyzed on the GC-MS by direct injection or after silylation as trimethylsilyl derivatives of the nonvolatile components. To obtain the trimethylsilyl derivatives, 500 µL of N-methyl-N-(trimethylsilyl) trifluoroacetamide (MSTFA) was added to the dried extracts and heated for 30 min at 70 °C. Thereafter, 500 µL of pyridine was added to the reaction, filtered, and transferred to GC-MS vials.

Gas chromatography–mass spectrophotometry analysis was carried on a Shimadzu QP 2100 SE (Shimadzu Corporation, Tokyo, Japan) equipped with an InertCap 5 MS/NP capillary (30 m × 0.25 mm × 0.25 μm: GL Sciences, Tokyo, Japan) capillary column. In total, 1 µL of each sample was injected into the GC-MS in split or split-less mode depending on the concentration of the extracts. Helium was used as the carrier gas at a constant flow rate of 1.0 mL/min. The oven temperature was programmed at 80 °C for 3 min, increased to 280 °C at 11 °C min^−1^, and then held at this temperature for 14 min. The analysis was carried out at 70 eV in the electron impact ionization mode. Compounds were identified tentatively based on comparison with published mass spectra libraries NIST 11 and Willey 10th edition and diagnostic ions.

### 4.3. Molecular Docking Studies

Molecular docking studies were conducted to determine the AQS potential and the mode of interactions of selected phytochemical compounds identified from *Melianthus comosus* against the CviR protein of *C. violaceum* ATCC 12472 (PDB: 3QP1) and *P. aeruginosa* ATCC 9721 (PDB: 2UV0), as described by [28]. The 2-dimensional structure of the phytochemical compounds from *Melianthus comosus* was obtained from the chemical library PubChem and drawn on Canvas 3.5 and exported to Maestro 11.5. The crystallized structures of the CviR and 2UV0 proteins of *C. violaceum* ATCC 12472 and *P. aeruginosa* ATCC 9721, and different ligands were obtained from the Protein Data Bank database (PDB). Prior to the docking experiments, chemically correct models of the ligands were generated using the ligprep of Schrodinger, and the receptor structure through a protein preparation wizard. Thereafter, docking was done using the Glide ligand docking module and Glide receptor for the grids. All docking calculations were performed using AutoDock 4.0 and Grids (Schrodinger, LLC, New York, NY, USA) for the prepared protein generated using the protein grid generation module. Further modifications included removal of water (H_2_O) and metals before optimizing the hydrogen bonds thus forcing minimization resulting in the generated scores mimicking the potential energy change when the protein and the compound come together based on hydrogen.

### 4.4. Bacterial Strain and Growth Conditions

A bacterial strain of *Pseudomonas aeruginosa* ATCC 9721 was purchased from Sigma-Aldrich (Johannesburg, South Africa). A wild-type strain producing a QS-controlled purple pigment violacein (*Chromobacterium violaceum* ATCC 12472) was used for the qualitative and quantitative determination of QS inhibition, which was kindly provided by the Centre for Microbial, Ecology and Genomics (CMEG), University of Pretoria, South Africa. Guanosine (Lot no: BCCB9660) and quercetin (Lot no: LRAB7760) compounds were also purchased from Sigma-Aldrich (Johannesburg, South Africa). The purchased bacteria were grown in their respective media and under incubation conditions for batch–batch reproducibility as described by the Clinical and Laboratory Standard Institute [55]. The active bacterial cultures were prepared in Mueller Hinton (MH) and Luria Bertani (LB) media and incubated at 37 and 30 °C for *P. aeruginosa* and *C. violaceum*, respectively. For the maintenance of the bacterial strains, glycerol stock cultures of each organism were prepared and kept at −80 °C until required. Prior to each assay, the bacterium was grown for 24 h at 30 and 37 °C on a respective agar plate. A single or two colonies were transferred to sterile distilled water to obtain an absorbance (OD_600nm_) of 0.1. This adjustment of cell suspension was performed to achieve 0.5 Mc Farland standard equivalent.

### 4.5. Antibacterial Activity Using a Microdilution Assay

The minimum inhibitory concentration (MIC) of crude extracts and compounds against *P. aeruginosa* was determined, using the broth dilution method on 96-microwell plates as previously described by Pauw and Eloff [56], with slight modifications. Briefly, 100 µL of Mueller Hinton broth (MHB) was transferred into every well and 100 µL of each plant extract (in triplicate) was transferred into wells in Row A of the microtiter plate together with the negative (1% dimethyl sulfoxide) and positive control (ciprofloxacin) at starting concentration of 0.01 mg/mL. Additionally, a blank (sterile MH broth) and standardized bacterium (control) were prepared by transferring 200 µL to the wells, respectively. Two-fold serial dilutions were performed, resulting in decreasing concentrations over the range of 6.25–0.048 mg/mL. Thereafter, 100 µL of the standardized bacterium was added into each well. After 24 h incubation at 37 °C, 40 µL of P-iodonitrotetrazolium (INT, 0.2 mg/mL) was added and incubated for a further 30 min to 1 h until the color of the solution becme pink. Bacterial growth inhibition (clear wells, no color change) was assessed visually and recorded. The MIC was recorded as the lowest concentration of the extract that inhibited bacterial growth.

### 4.6. Evaluation of Plant Extracts for Anti-Quorum Sensing (AQS) Potential

#### 4.6.1. Qualitative Anti-Quorum Sensing Assay

Disc diffusion assay was carried out to detect the AQS activity of the crude extracts according to Chenia [21], with slight modification as follows. Briefly, the bacterium was grown for 24 h at 30 °C on LB agar plate and the bacterial suspension was adjusted to an OD_600nm_ of 0.1. The standardized culture was swabbed evenly on the agar plate surface. The sterile discs (6 mm diameter) were impregnated with 10 µL of crude extracts with varying concentrations MIC—1/8 MIC in mg/mL), with positive control of cinnamaldehyde (MIC—1/8 MIC in mg/mL) and incubated for 24 h at 30 °C. The plates were examined for violacein production, whereby the loss of purple violacein pigment (opaque zone) surrounding the disc indicated AQS while clear and transparent zones indicated bactericidal activity around the discs. The diameter of the clear/halo inhibition zones was interpreted as follows: Susceptible (S) ≥ 20 mm, Intermediate (I) = 15–19 mm and Resistant ≤ 14 mm, as described by CLSI [55].

#### 4.6.2. Quantitative Anti-Quorum Sensing Assay

Anti-quorum sensing activity of plant extracts and compounds was tested against the bacterium *C. violaceum* ATCC 12472 using the microdilution method described in Section 2.3, with cinnamaldehyde used as positive control. Before incubation, the absorbance was read at OD_600nm_ (to check the viability and growth of the bacterium) and OD_485nm_ (violacein production). The plates were then incubated at 30 °C for 24 h, shaking at 120 rpm. Following incubation, absorbance was read again at O42_0nm_. Thereafter the plates were placed in a drying oven at 50 °C for 24 h. After drying, to confirm that plant extracts inhibit quorum sensing without influence on bacterial growth activity, 150 µL of 100% DMSO was used to re-suspend the dried contents in each well, mixed thoroughly and placed in the shaking incubator at 30 °C, 120 rpm for 1–2 h(s). Thereafter, absorbance was read at an OD_485nm_ for violacein quantification. The percentage (%) inhibition was determined using Equation (2):Percentage (%) inhibition = (OD control−OD test)/(OD control) × 100(2)
where OD is the optical density taken at an absorbance of 485 nm.

### 4.7. Effect of Plant Extracts and Selected Compounds on Cell Attachment and Biofilm Development

Cell attachment (antiadhesion) and biofilm growth (mature biofilm) were assessed for inhibition using the crude extracts and compounds, this method was followed according to Famuyide et al. [57] with slight modifications. Briefly, the three *M. comosus* (aqueous, methanol and dichloromethane) extracts with noteworthy MIC values (≤ 1 mg/mL) were tested against *P. aeruginosa* for both cell attachment and biofilm development inhibition. In the cell attachment inhibition assay, 100 μL of standardized bacterial suspension (OD_600nm_ = 0.1), 100 μL of MH broth and 100 μL of extract were added to the wells. The positive control (ciprofloxacin, 0.001 mg/mL) and negative control (1% DMSO) was also added into the wells. The blank wells were added with 200 μL of sterile MH broth, thereafter, incubated at 37 °C for 24 h.

For biofilm development bioassays, 100 μL of standardized bacterial suspension and 100 μL of MH broth was added to the wells and incubated at 37 °C for 8 h. After incubation, 100 μL of extracts and controls were transferred into respective wells and incubated further for 24 h. Biofilm biomass was assessed using the modified crystal violet (CV) assay. The 96-well plates containing formed biofilm were washed with sterile distilled water to remove planktonic cells and media. The plates were then oven-dried at 60 °C for 45 min. Following drying, 1% CV solution (Sigma-Aldrich, Johannesburg, South Africa) was used to stain the remaining biofilm for 15 min in the dark. The wells were then washed with sterile distilled water to remove any unabsorbed stain. Semiquantitative assessment of biofilm formation was performed by adding 125 µL of 95% ethanol to destain the wells. In total, 100 µL of the destaining solution was transferred to a new plate and the absorbance (OD 585 nm) was determined using a multimode microplate reader (SpectraMax^®^ paradigm). The percentage of inhibition was determined using equation 2.

The following criterion for interpretation of results was used; whereby values between 0 and 100% were interpreted as an inhibitory activity, then further, breaking it down as follows: ≥ 50% (good activity), values between 0 and 49% (weak activity) while negative values reflect the enhancement of growth instead of biofilm inhibition [57].

### 4.8. Inhibition of Quorum Sensing Mediated Virulence Determinants—Pyocyanin Assay

The pyocyanin assay was performed according to the method described by Bhattacharya et al. [58], with slight modifications. Briefly, an overnight culture of *P. aeruginosa* was diluted to OD_600nm_ of 0.1. Thereafter, plant extracts, and guanosine were added with different concentrations (2 x MIC—¼ MIC in mg/mL), including the standardized culture in King’s A broth and incubated overnight at 37 °C. A volume of 1.5 mL of overnight culture was centrifuged at 3000× *g* for 10 min. Afterwards, 1 mL of supernatant was transferred into fresh centrifuge tubes (pre-cooled in ice), allowed to chill and 100 μL chloroform was added while in ice. Then, 300 μL of 0.2 M hydrochloric acid (HCl) was added and mixed vigorously using vortex mixer (EINS Sci E-VM-A, Biotechnology Hub Africa, Hatfield, South Africa). The chloroform layer containing pyocyanin was collected and transferred into a 96-wells microtiter plate. The absorbance was read at 520 nm using microtiter plate reader (SpectraMax^®^ paradigm). The experiments were performed in triplicates to get the mean value. Ciprofloxacin (0.001 mg/mL) and 1% DMSO were used as the positive and negative controls, respectively. Pyocyanin concentration was calculated by multiplying the OD value at 520 nm with 17.072 (the molar extinction coefficient). Pyocyanin production was compared with untreated cells, used as a control.

### 4.9. Swimming and Swarming Motility Assay

The motility assay was performed following the method described by Cosa et al. [3], with slight modifications. Swimming media consisted of 1% tryptone, 0.5% NaCl, and 0.5% agar. The medium prepared for the swarming assay was composed of nutrient broth (0.8%, *w*/*v*), supplemented with glucose (3%, *w*/*v*) and agar of 0.5% (*w*/*v*). The standardized bacterium (2 µL) was spotted on agar plates containing swimming and swarming media supplemented with or without extract. Each plant extract was tested at varying concentrations (2 x MIC—¼ MIC in mg/mL) as well as guanosine (2 x MIC—¼ MIC in mg/mL). Ciprofloxacin and 1% DMSO were used as positive and negative controls, respectively. The plates were incubated at 37 °C for 24 h. The zone diameters (mm) were measured to assess swimming and swarming motility and compared to the negative and positive controls. The experiments were performed in triplicates, to obtain the mean value.

### 4.10. Statistical Analysis

All results were presented as mean ± standard deviations for each sample and treatments carried out in triplicates. Means from inhibitory activities of extracts and controls were analyzed using ANOVA generalized linear model (Proc GLM). Means were separated using least significant difference (LSD) method. All statistical analyses were performed using a Statistical Analysis System (SAS) program version 9.4, Stats Inc., 100 SAS Campus Drive, Cary, NC, USA, and *p* < 0.05 values were considered statistically significant.

## 5. Conclusions

The findings of this study highlighted the potential of *Melianthus comosus* extracts and guanosine in reduction of QS-dependent virulence factors in *P. aeruginosa* and presented the significance of molecular docking strategy in the discovery of novel compounds. The plant extracts showed the potential to inhibit *P. aeruginosa* bacterium and disrupt the violacein production of *Chromobacterium violaceum*. Guanosine, a compound present in *Melianthus comosus* revealed potent QSI results in *P. aeruginosa* and *C. violaceum*.

Based on the results of the in vitro studies, the Las quorum sensing system in *P. aeruginosa* as well as the Rhl system were targeted while the PQS system and the IQS system were not. Compounds of sucrose, guanosine and 1,2-benzene dicarboxylic acid, diethyl ester were shown as good quorum quenching compounds as they bound to either or both QS regulatory proteins of CviR (3QP1) and LasR (2UV0). Future studies will focus on experiments to validate the antipathogenic potential of these compounds, in their pure form. Plant extracts and their antivirulence compounds are promising in the development of new drugs that can circumvent resistant *P. aeruginosa*.

## Figures and Tables

**Figure 1 antibiotics-10-00679-f001:**
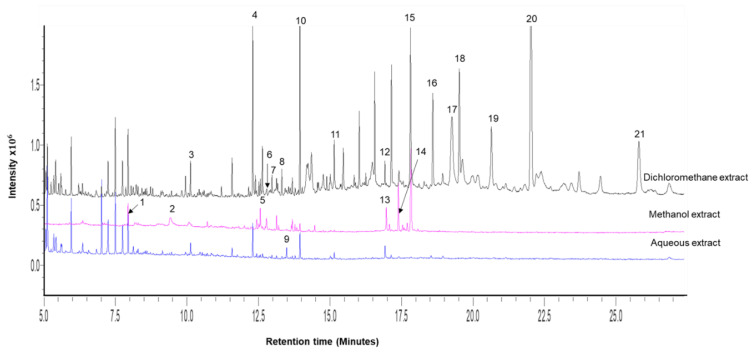
Representative total ion chromatograph (TIC) of *Melianthus comosus* (aqueous, methanol and dichloromethane) extracts. All peaks correspond to the data presented in Table 1.

**Figure 2 antibiotics-10-00679-f002:**
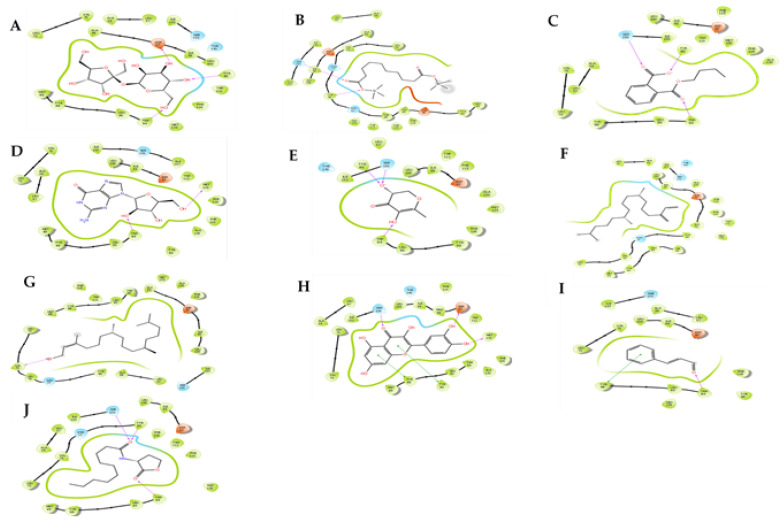
Interaction network between 3QP1 protein and the studied compounds. The protein residues with a negative charge are shown in red, positive charge in velvet, polar in cyan, and hydrophobic in parrot green. The H-bond interactions are shown as a purple arrow, pi-pi stacking as a green line. (**A**) Sucrose; (**B**) decanedioic acid; (**C**) 1,2-benzenedicarboxylic acid; (**D**) guanosine; (**E**) 2.3-dihydro-3.5-dihydroxy-6-methyl-4H-pyran-4-one; (**F**) neophytadiene; (**G**) phytol; (**H**) quercetin; (**I**) cinnamaldehyde and (**J**) C10-homoserinelactone.

**Figure 3 antibiotics-10-00679-f003:**
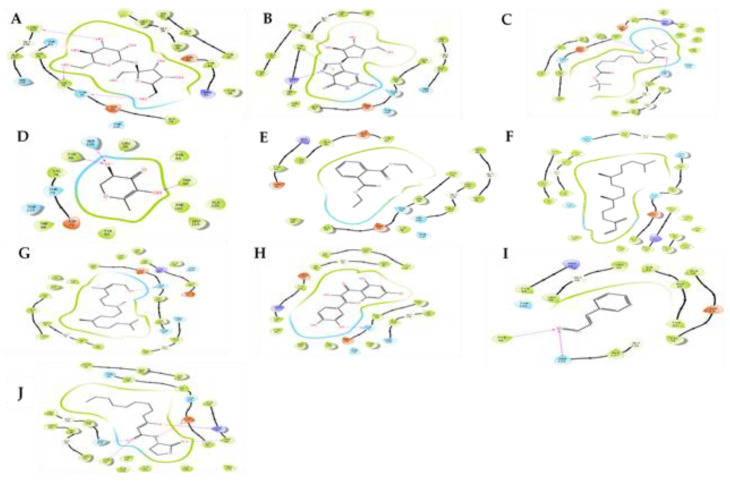
Interaction network between 2UV0 protein and the studied compounds. The protein residues with a negative charge are shown in red, positive charge in velvet, polar in cyan, and hydrophobic in parrot green. The H-bond interactions are shown as a purple arrow, pi-pi stacking as a green line. (**A**) Sucrose; (**B**) guanosine; (**C**) azelaic acid; (**D**) 2.3-dihydro-3.5-dihydroxy-6-methyl-4H-pyran-4-one; (**E**) 1,2-benzenedicarboxylic acid, diethyl ester; (**F**) neophytadiene; (**G**) phytol; (**H**) quercetin; (**I**) cinnamaldehyde and (**J**) 3-oxo-C12-HSL.

**Figure 4 antibiotics-10-00679-f004:**
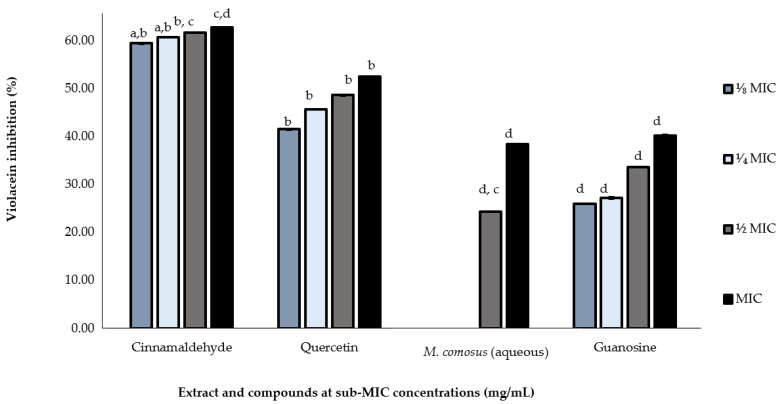
Quantitative violacein inhibition (%) using *Melianthus comosus* (aqueous) extract and guanosine against *C. violaceum* at concentrations between 0.190 and 1.56 mg/mL (MIC–⅟₈ MIC). Data are represented as the percentage of violacein inhibition. Control (untreated) was set as 100% production. Mean are values of triplicate independent experiments ± SD. Comparison for each concentration across the different individual treatments, presented with different letters (**a**–**d**) indicate significant difference at *p* < 0.05.

**Figure 5 antibiotics-10-00679-f005:**
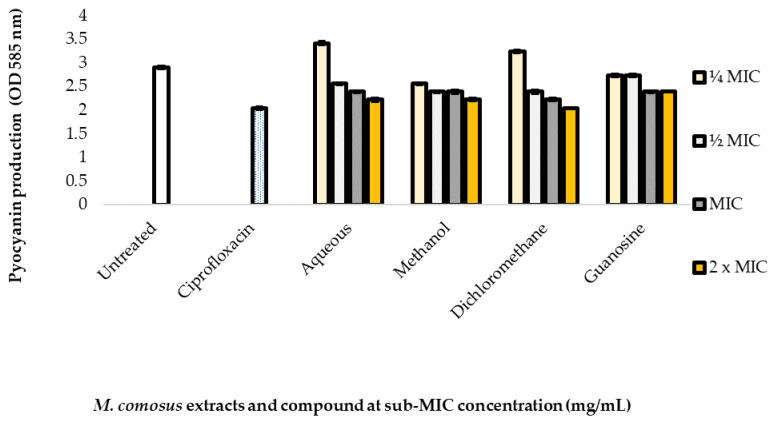
Pyocyanin production inhibition using *Melianthus comosus* (aqueous, methanol and dichloromethane) extracts and guanosine against *P. aeruginosa* at concentration between 1.56 and 0.195 mg/mL. Data are represented as OD values read at absorbance of 585 nm. Mean are values of triplicate independent experiments ± SD. Untreated was set as control for *P. aeruginosa* pyocyanin production.

**Figure 6 antibiotics-10-00679-f006:**
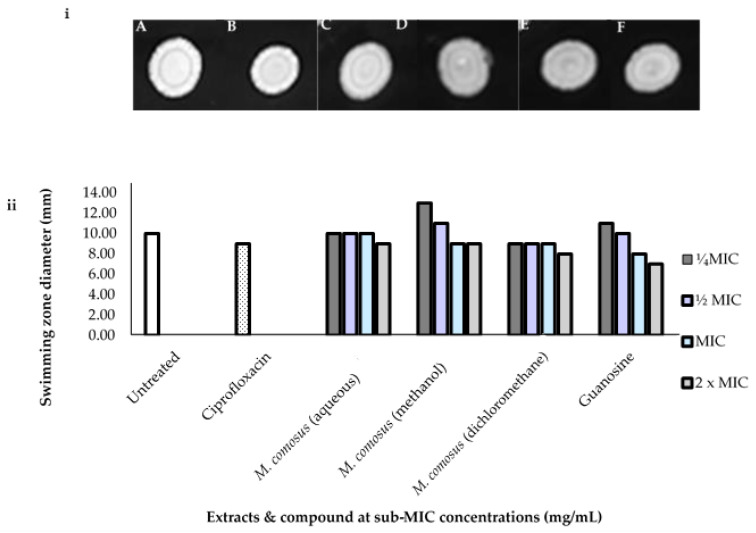
(**i**) (**A**) Negative control (untreated cells); (**B**) ciprofloxacin, (**C**) *M. comosus* (aqueous), (**D**) *M. comosus* (methanol) (**E**) *M. comosus* (dichloromethane) extracts, respectively, and (**F**) guanosine are representative images of *P. aeruginosa* swimming motility treated with *M. comosus* (aqueous, methanol and dichloromethane) extracts and guanosine at 1.56 and 0.062 mg/mL (MIC × 2) concentration, respectively. (**ii**) Bar graph showing *M. comosus* extracts and guanosine at sub-MIC concentrations. Mean values are of triplicate zone diameters.

**Figure 7 antibiotics-10-00679-f007:**
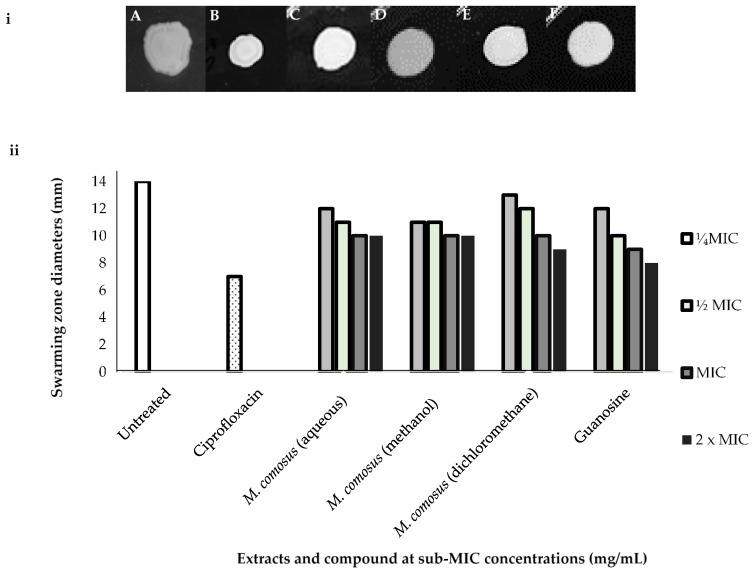
(**i**) (**A**) Negative control (untreated cells), (**B**) ciprofloxacin, (**C**) *M. comosus* (aqueous), (**D**) *M. comosus* (methanol), (**E**) *M. comosus* (dichloromethane) extracts and (**F**) guanosine are representative images of *P. aeruginosa* swarming motility treated with *M. comosus* (aqueous, methanol and dichloromethane) extracts and guanosine at 1.56 and 0.062 mg/mL (2 x MIC) concentration, respectively, and (**ii**) bar graph showing *M. comosus* extracts and guanosine at sub-MIC concentrations. Mean values are of triplicate zone diameters.

**Table 1 antibiotics-10-00679-t001:** GC-MS spectral analysis of *Melianthus comosus* (aqueous, methanol and dichloromethane) extracts.

Peak #	Ret. Time (min)	Name	Molecular Weight	Molecular Formula	*M. comosus* Extracts		
					AQ	ME	DCM
1	4.453	Decane	142	C_10_H_22_	14.1%	11.3%	
2	6.479	2,3-Dihydro-3,5-dihydroxy-6-methyl-4H-pyran-4-one	144	C_6_H_8_O_4_	5.0%		
3	7.815	Cyclooctane *****	112	C_8_H_16_		1.2%	
4	9.389	Guanosine	283	C_10_H_13_N_4_O	9.8%	22.3%	
5	10.07	Propanoic acid *****	174	C_8_H_18_O_2_Si		1.4%	
6	10.837	1,2-Benzenedicarboxylic acid, diethyl ester	236	C_12_H_14_NO_6_	64.1%	52.8%	2.0%
7	11.576	Octanedioic acid *****	318	C_14_H_30_O_4_Si_2_	1.2%		1.2%
8	12.267	(-)-Loliolide	196	C_11_H_16_O_3_			0.2%
9	12.298	Azelaic acid *****	332	C_15_H_32_O_4_Si_2_	4.8%		5.6%
10	12.44	D-Galactose, 2,3,4,5,6-pentakis-O *****	540	C_21_H_52_O_6_Si_5_	0.4%	2.5%	
11	12.512	D- (-)-Fructofuranose, pentakis ether *****	541	C_21_H_52_O_6_Si_5_		2.0%	0.5%
12	12.597	Neophytadiene	278	C_20_H_38_	2.4%	5.4%	0.7%
13	12.638	Tetradecanoic acid *****	300	C_17_H_36_O_2_Si			1.4%
14	12.894	Cyclopentadecanol	226	C_15_H_30_O		2.2%	0.4%
15	12.945	Decanedioic acid *****	346	C_16_H_34_O_4_Si_2_			0.45%
16	13.483	Benzoic acid 3,4,5-tris(trimethylsiloxy) *****	458	C_19_H_38_O_5_Si_4_	1.5%		
18	13.949	Hexadecanoic acid *****	328	C_19_H_40_O_2_Si	3.4%	1.4%	5.4%
19	14.407	Tetramethyl hexadecenol (Trans phytol)	296	C_20_H_40_O		4.8%	0.9%
20	14.89	Docosane	310	C_22_H_46_			0.4%
21	15.125	Octadecanoic acid *****	356	C_21_H_44_O2Si			1.4%
22	15.47	Tricosane	324	C_23_H_48_			1.2%
23	16.025	Tetracosane	338	C_24_H_50_			2.5%
24	16.562	Pentacosane	352	C_25_H_52_			4.2%
25	16.919	1,2-Benzenedicarboxylic acid, mono(2-ethylhexyl) ester *****	390	C_24_H_38_O_4_	1.9%		0.7%
26	16.969	Sucrose *****	342	C_12_H_22_O_11_		4.8%	
27	17.146	Hexacosane	366	C_26_H_54_			5.7%
28	17.7	Thymol-. beta. -d-glucopyranoside, tetrakis *****	600	C_28_H_56_O_6_Si_4_		2.2%	
29	17.808	Heptacosane	380	C_27_H_56_			7.0%
30	17.814	Tetratetracontane *****	618	C_44_H_90_			6.9%
31	18.584	Octacosane	394	C_28_H_58_			6.4%
32	19.505	Nonacosane	408	C_29_H_60_			9.6%
33	19.262	1-Hentetracontanol *****	592	C_41_H_84_O			7.2%
34	20.625	Triacontane	422	C_30_H_62_			5.0%
35	22.00	Dotriacontane	450	C_32_H_66_	4.6%		33.9%
36	22.03	Hexatraicontane *****	506	C_36_H_74_			21.6%
37	22.86	alpha-Tocopherol (Vitamin E)	430	C_29_H_50_O_2_			1.6%
38	23.66	Tetratricontane	478	C_34_H_70_			3.0%
39	26.7	Stigmast-5-en-3-ol, (3. beta.)	414	C_29_H_50_O		2.7%	4.6%

***** Indicates compounds that are derivates. AQ: aqueous; ME: methanol and DCM: dichloromethane.

**Table 2 antibiotics-10-00679-t002:** *Melianthus comosus* crude extract including guanosine and their minimum inhibitory concentrations (MIC) against *Pseudomonas aeruginosa*.

*Melianthus comosus* Extracts	MIC (mg/mL)
Aqueous	0.78
Methanol	0.78
Acetone	1.56
Ethyl acetate	6.25
Dichloromethane	0.78
Compounds	
Guanosine	0.031
1% DMSO	≥6.25
Ciprofloxacin	0.001
Quercetin	0.008

**Table 3 antibiotics-10-00679-t003:** Inhibitory (IC_50_) values of the most active plant extract and compounds against *C. violaceum*.

Plant Species	Half Y	IC_50_ (mg/mL)
*Melianthus comosus* (aqueous)	37.66	1.52
Compounds		
Cinnamaldehyde	56.35	0.087
Quercetin	33.44	0.043
Guanosine	21.05	0.064

**Table 4 antibiotics-10-00679-t004:** Percentage inhibition of cell attachment and biofilm development by *P. aeruginosa*, following exposure to aqueous, methanolic and dichloromethane extracts of *M. comosus*.

*M. comosus* Extracts	Cell Attachment (%)	Biofilm Development (%)
Aqueous	−47.91 ± 0.51 ^a^	−57.21 ± 0.34 ^b^
Methanol	−10.17 ± 0.12 ^b^	−48.54 ± 0.45 ^a,b^
Dichloromethane	−14.25 ± 0.46 ^b^	−128.36 ± 0.59 ^a^
Compounds		
Guanosine	78.88 ± 0.11 ^d^	34.85 ± 1.08 ^c^
Ciprofloxacin	52.88 ± 0.18 ^c^	39.43 ± 0.30 ^c^
Quercetin	55.14 ± 0.03 ^c^	44.35 ± 0.05 ^c,d^

Mean values are of triplicate independent experiments ± SD. Comparison of percentage inhibition at MIC value for each treatment against *P. aeruginosa*. Different letters (a–d) indicate significant difference at *p* < 0.05.

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
