# Peer review of "In Silico and In Vitro Screening of Antipathogenic Properties of Melianthus comosus (Vahl) against Pseudomonas aeruginosa"

_antibiotics, 2021, doi:10.3390/antibiotics10060679_

Round 1
Reviewer 1 Report
In this study, the authors proposed that plant extracts of Melianthus comosus can inhibit the quorum sensing (QS) system of Pseudomonas aeruginosa. The component analysis of the extracts was carried out by GC-MS and in silico evaluation narrowed down the candidates of active compounds. In these candidates, guanosines was focused and evaluated their QS inhibitory activity. The ability of these plant extracts to reduce pathogenicity indicators such as community motility and biofilm formation in P. aeruginosa is an interesting finding, and the findings of this study can be generally accepted. The authors should consider revising the following points for further improvement. (1) In silico experiments, by themselves, do not provide clear evidence of interaction with target molecules. The statements about Docking Score and energy given in Tables 2 and 3 are acceptable. However, the discussion of the interaction mode based on the prediction of the amino acid residues on the receptors illustrated in Figures 2 and 3 and also in the tables and text is an overestimation of the prediction. In my opinion, this part should be deleted and described concisely. (2) The representative data in Table 7 should be shown in graphs and photos which are visually understandable. (3) Discussion is too long for the content. Authors should revise it more concise.Author Response
COMMENTS FROM REVIEWER 1.
Reviewer 1 (general comment):
In this study, the authors proposed that plant extracts of Melianthus comosus can inhibit the quorum sensing (QS) system of Pseudomonas aeruginosa. The component analysis of the extracts was carried out by GC-MS and in silico evaluation narrowed down the candidates of active compounds. In these candidates, guanosine was focused and evaluated their QS inhibitory activity. The ability of these plant extracts to reduce pathogenicity indicators such as community motility and biofilm formation in P. aeruginosa is an interesting finding, and the findings of this study can be generally accepted. The authors should consider revising the following points for further improvement.
Response: Thank you for the time and effort in reviewing our manuscript. We acknowledge that it would improve the revised version of the manuscript.
First comment:
In silico experiments, by themselves, do not provide clear evidence of interaction with target molecules. The statements about Docking Score and energy given in Tables 2 and 3 are acceptable. However, the discussion of the interaction mode based on the prediction of the amino acid residues on the receptors
illustrated in Figures 2 and 3 and also in the tables and text is an overestimation of the prediction. In my opinion, this part should be deleted and described concisely.
Response: We thank you for the comment and truly appreciate the suggestion. We agree with the reviewer only on the repetitive presentation of the data in the table and on the figures. The interactions have been briefly described. However, authors would rather present Figures 7 and 8 on basis that the interpretation and interactions of the amino acids and hydrogen bonding was done, supported by documented literature [Kumar et al., 2015 and Singh and Bhatie, 2018], referenced in this study as [2 and 30, respectively]. Further, the software Autodock vina that was used in the present study for molecular docking provides accurate and reliable data on receptor and ligand interactions. The figures demonstrate the interactions of the amino acids on the receptor protein against the ligands (test compounds). The figure legends state how the interactions are interpreted. The interactions are also compared to that of the AHL molecule native to this receptor site, and the quercetin (positive control) binding to this receptor site.
Second comment: The representative data in Table 7 should be shown in graphs and photos which are visually understandable.
Response: Thank you for the suggestion. For pyocyanin production results, we have presented results in Figure 3, on page 13. Swimming results are presented in Figure 4 and showed representative images of the zone diameters, on page 14. Swarming results are also presented in Figure 5 and showed representative images of the zone diameters, on page 15.
Third comment:
The discussion has been revised; details are explained in the table below:
|
Reviewer 1 Comments: |
Author response: |
|
Discussion is too long for the content. Authors should revise it more concise |
Thank you for the suggestion. The following paragraphs in discussion section has been deleted: Page 18, line 501-503 and line 524 - 526; Page 19 line 551 and line 577; Page 20, line 591-593 and line 603 - 613; Page 21, line 681 - 682; Page 22, line 697-699 and line 723-726 |
|
In reference section, the following has been deleted, as a result of the revised discussion. [27] – Ododo et al., 2016 [28] – Gołezbiowski et al., 2013 [42] – Skogman et al., 2016 [43] – Malešević et al., 2019 |
Thank you
Reviewer 2 Report
Dear authors,
The article follows a realiable and valid experimental approach. All the tests have been correctly done and only some minor corrections are needed before publication. I suggest an English revision because some parts are not clear.
Comments and suggested changes are reported on the pdf article.

Author Response
COMMENTS FROM REVIEWER 2.
Reviewer 2 (general comment):
The article follows a reliable and valid experimental approach. All the tests have been correctly done and only some minor corrections are needed before publication. I suggest an English revision because some parts are not clear.
Response: Thank you for the time and effort in reviewing our manuscript. We acknowledge that it would improve the revised version of the manuscript.
|
Reviewer 2 Comments: |
Author response: |
|
On page 1, line 2. The reviewer removed for and added of |
On page 1, line 2. In accordance with the reviewer’s suggestion, we have now changed “for” to “of” |
|
On page 1, line 16. Reviewer suggested we “add three” |
On page, line 16. In accordance with the reviewer’s suggestion, we have added ‘three’ |
|
On page 1, line 20. Reviewer asked, “Is this value referred to the common compounds?” Please specify that they have obtained the same MIC value |
On page 1, line 20. The 0.78 mg/mL is the MIC value of the three crude extracts. I have also added “the same MIC value” as reviewer suggested. |
|
On page 1, line 20. Reviewer asked “both?” |
On page 1, line 20. It is for all three crude extracts |
|
On page 1, line 21. Reviewer asked, “why is it called "selected"?”
|
On page 1, line 21. Because guanosine was the best quorum sensing potential compound, binding to both 3QP1 and 2UV0 receptor protein, hence it was selected to be investigated further for in-vitro studies. |
|
On page 1, line 22. Reviewer asked and suggested. “What does AQS stand for?” Specify the acronym, especially for the letter A, and remember to do this at the beginning of every section (abstract, introduction, etc). |
On page 1, line 22. In accordance with the reviewer’s suggestion, AQS is now written in full. It stands for ‘Anti-quorum sensing’ |
|
On page 2, line 53. Reviewer suggested: Quorum sensing S? specify the acronym the first time of appearance |
On page 2, line 53. In accordance with the reviewer’s suggestion, QSS is now written in full |
|
On page 3, line 108. Reviewer suggested “correct the name” |
On page 3, line 108. In accordance with the reviewer’s suggestion, the name is corrected. Now written as: Chromobacterium |
|
On page 3, line 115. Reviewer suggested to use 'extracts' instead of 'extracted' |
On page 3, line 115. In accordance with the reviewer’s suggestion, extracts have been changed to ‘extracted’ |
|
On page 3, line 128. Reviewer asked, “Are you referring to the compound present in the extracts?” Please specify |
On page 3, line 128. We have provided clarity on about the compounds. Yes, we were referring to the compounds present in the extracts. |
|
On page 5, Table 1. Reviewer asked, “Please add a note with the corresponding name of acronyms” |
On page 5, Table 1. We have added corresponding names to the acronyms below the table. |
|
On page 9, line 210. Reviewer deleted “when” |
On page 9, line 210. We have deleted “when”. |
|
On page 9, line 210. Reviewer deleted “in” |
On page 9, line 210. We have deleted “in”. |
|
On page 9, line 211. Reviewer deleted “this was performed” and suggested “in order to” |
On page 9, line 211. We have deleted “this was performed” and added “in order to”. |
|
On page 12, line 258. Reviewer suggested “Maybe you can say...showed the SAME MIC value...” |
On page 12, line 258. We added “showed the same”. |
|
On page 12, Table 4 Reviewer asked, “DMSO is a control?” |
On page 12, line 260, we have included a sentence to specify that DMSO was used as a negative control. |
|
On page 12, line 281. Reviewer asked “which concentrations? Specify” |
On page 12, line 281. We have included the concentrations. |
|
On page 14, Table 6. Reviewer asked “please, specify that this is a percentage value. Moreover, do positive values correspond to inhibition of biofilm development? And negative the contrary?”
|
On page 14, Table 6. We have added the percentage value as suggested.
Yes, positive values indicate inhibition of biofilm formation while the negative value indicates no inhibition. |
|
On page 19, line 527. Reviewer corrected violecein to violacein |
On page 19, line 527. We have corrected violecein to violacein. |
|
On page 24, line 766. Reviewer asked, “at which OD? only visually?”
|
We assessed the MIC values only visually. For future studies, we will also use a microtiter plate reader to get OD values and determine the MIC values. |
Concluding Remarks:
Once again, thank you for giving us the opportunity to strengthen our manuscript with your valuable comments and suggestions. We look forward to hearing from you regarding our submission. We would be glad to respond to any further questions and comments that you may have.
Round 2
Reviewer 1 Report
Most of the points raised have been addressed. I agree to the publication of this article.